**Key words:**
Cryo-EM; human coronavirus NL63; spike trimer

**\*Author for correspondence:**
Wah Chiu, E-mail: wahc@stanford.edu

# A 3.4-Å cryo-electron microscopy structure of the human coronavirus spike trimer computationally derived from vitrified NL63 virus particles

Kaiming Zhang[1] ⓘ, Shanshan Li[1] ⓘ, Grigore Pintilie[1] ⓘ, David Chmielewski[2] ⓘ, Michael F. Schmid[4] ⓘ, Graham Simmons[3] ⓘ, Jing Jin[3,4] ⓘ and Wah Chiu[1,4]* ⓘ

[1]Department of Bioengineering and James H. Clark Center, Stanford University, Stanford, CA 94305, USA; [2]Graduate Program in Biophysics, Stanford University, Stanford, CA 94305, USA; [3]Vitalant Research Institute, San Francisco, CA 94030, USA and [4]Division of CryoEM and Bioimaging, SSRL, SLAC National Accelerator Laboratory, Menlo Park, CA 94025, USA

**Abstract**

Human coronavirus NL63 (HCoV-NL63) is an enveloped pathogen of the family *Coronaviridae* that spreads worldwide and causes up to 10% of all annual respiratory diseases. HCoV-NL63 is typically associated with mild upper respiratory symptoms in children, elderly and immuno-compromised individuals. It has also been shown to cause severe lower respiratory illness. NL63 shares ACE2 as a receptor for viral entry with SARS-CoV-1 and SARS-CoV-2. Here, we present the *in situ* structure of HCoV-NL63 spike (S) trimer at 3.4-Å resolution by single-particle cryo-EM imaging of vitrified virions without chemical fixative. It is structurally homologous to that obtained previously from the biochemically purified ectodomain of HCoV-NL63 S trimer, which displays a three-fold symmetric trimer in a single conformation. In addition to previously proposed and observed glycosylation sites, our map shows density at other sites, as well as different glycan structures. The domain arrangement within a protomer is strikingly different from that of the SARS-CoV-2 S and may explain their different requirements for activating binding to the receptor. This structure provides the basis for future studies of spike proteins with receptors, antibodies or drugs, in the native state of the coronavirus particles.

## Introduction

*Coronaviridae* constitute a large family of enveloped, positive-sense single-stranded RNA (+ssRNA) viruses capable of causing severe and widespread human respiratory disease. The coronaviruses are zoonotic pathogens, often circulating among natural reservoirs, such as bats or camels prior to crossing the species barrier into humans (Wang *et al.,* 2006; Sabir *et al.,* 2015). Coronaviruses are classified into four genera (alpha-CoV, beta-CoV, gamma-CoV and delta-CoV), with human coronaviruses found in two: alpha-CoVs (HCoV-229E and HCoV-NL63) and beta-CoVs [middle-east respiratory syndrome (MERS) and severe acute respiratory syndrome coronavirus 1 and 2 (SARS-CoV-1 and SARS-CoV-2)] (Cui *et al.,* 2018). SARS-CoV-2, responsible for the COVID-19 pandemic, has infected over 47 million people and claimed over 1.2 million lives worldwide as of early November 2020, underscoring the urgency of studying all circulating human coronaviruses (https://covid19.who.int/) (Cui *et al.,* 2018; Zhou *et al.,* 2020). While beta-CoVs are more commonly associated with high pathogenicity and severe respiratory disease, alpha-CoVs are widely circulating viruses associated with cold-like symptoms, and in rare cases, more serious respiratory failure (Wang *et al.,* 2020). HCoV-NL63 is estimated to be the causative agent of up to 10% of annual respiratory disease and is a major cause of bronchiolitis and pneumonia in newborns (van der Hoek *et al.,* 2004; Chiu *et al.,* 2005).

Coronaviruses utilize large spike (S) homotrimers (500-600 kDa) protruding from the viral membrane to engage cellular receptors and mediate fusion with host membranes (Thorp *et al.,* 2006; Pallesen *et al.,* 2017). These spikes, numerous on the virus surface, constitute the primary target of neutralizing antibodies (NAbs) and are central to vaccine design and structural studies for drug optimization. Each S protomer contains two large regions: N-terminal S1 responsible for receptor-binding and C-terminal S2 responsible for type-I fusion, as well as an additional single-pass transmembrane helix that anchors the spike to the viral envelope (Bosch *et al.,* 2003; Zheng *et al.,* 2006). Spikes are activated by protease cleavage, near the putative fusion peptide, allowing the transition from pre- to post-fusion conformation during virus entry (Belouzard *et al.,* 2009).

Several single-particle cryo-electron microscopy (cryo-EM) structures of purified ectodomains of pre-fusion S trimers from alpha-CoV and beta-CoV genera have been determined,

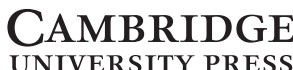

illustrating conformational heterogeneity of S1, multiple glycosylation sites, and the interactions with soluble receptors and NAbs (Walls *et al.,* 2016, 2020; Song *et al.,* 2018; Wrapp *et al.,* 2020). These S complexes lack the helical C-terminal stem that connects S2 to the viral envelope, and require residues mutated to proline at the loop between the first heptad repeat and the central helix to stabilize the construct (Pallesen *et al.,* 2017; Kirchdoerfer *et al.,* 2018; Park *et al.,* 2019). High-resolution structure determinations of molecular components in pleomorphic virions are typically obtained by single-particle cryo-EM or X-ray crystallography of exogenously expressed and purified components. Conversely, cryo-electron tomography (cryo-ET) has been used to reconstruct tomograms of the whole virus particles, from which sub-volume density, corresponding to the feature of interest, would be further processed by classification and averaging (Schmid *et al.,* 2012; Wan and Briggs, 2016). Here, we take the approach of utilizing single-particle cryo-EM imaging of purified and vitrified HCoV-NL63 virus particles, and then using computational methods to extract the S trimer density for its high-resolution structure determination. Our approach is similar to a recent preprint report for chemically fixed SARS-CoV-2 particles (Ke *et al.,* 2020). We report the 3.4-Å

structure of the S trimer in its native, pre-fusion state, and expect our data will be of value in future drug and vaccine design for various strains of coronaviruses without the need to genetically construct and biochemically purify spike proteins.

## Results and discussion

Since the S trimers protrude from the membrane surface of the virion, many of the crown-shaped S trimers can be easily identified, computationally extracted from the raw images, and treated as single-particle images with a distinct particle orientation without the need of tilting the specimens (Fig. 1*a*). Based on the fact that single-particle cryo-EM is a more routine procedure for computing high-resolution structures compared to cryo-ET, we decided not to collect tilt series, and used single-particle cryo-EM analysis of the native S trimer on purified and vitrified virions. The two-dimensional (2D) class averages showed a clear triangular shape for the S trimer and other distinct views (Fig. 1*b*). *De novo* building of the initial map, using the 'Ab-initio reconstruction' option in cryoSPARC without any symmetry applied, resulted in a three-dimensional (3D) structure with well-defined features. Further 3D

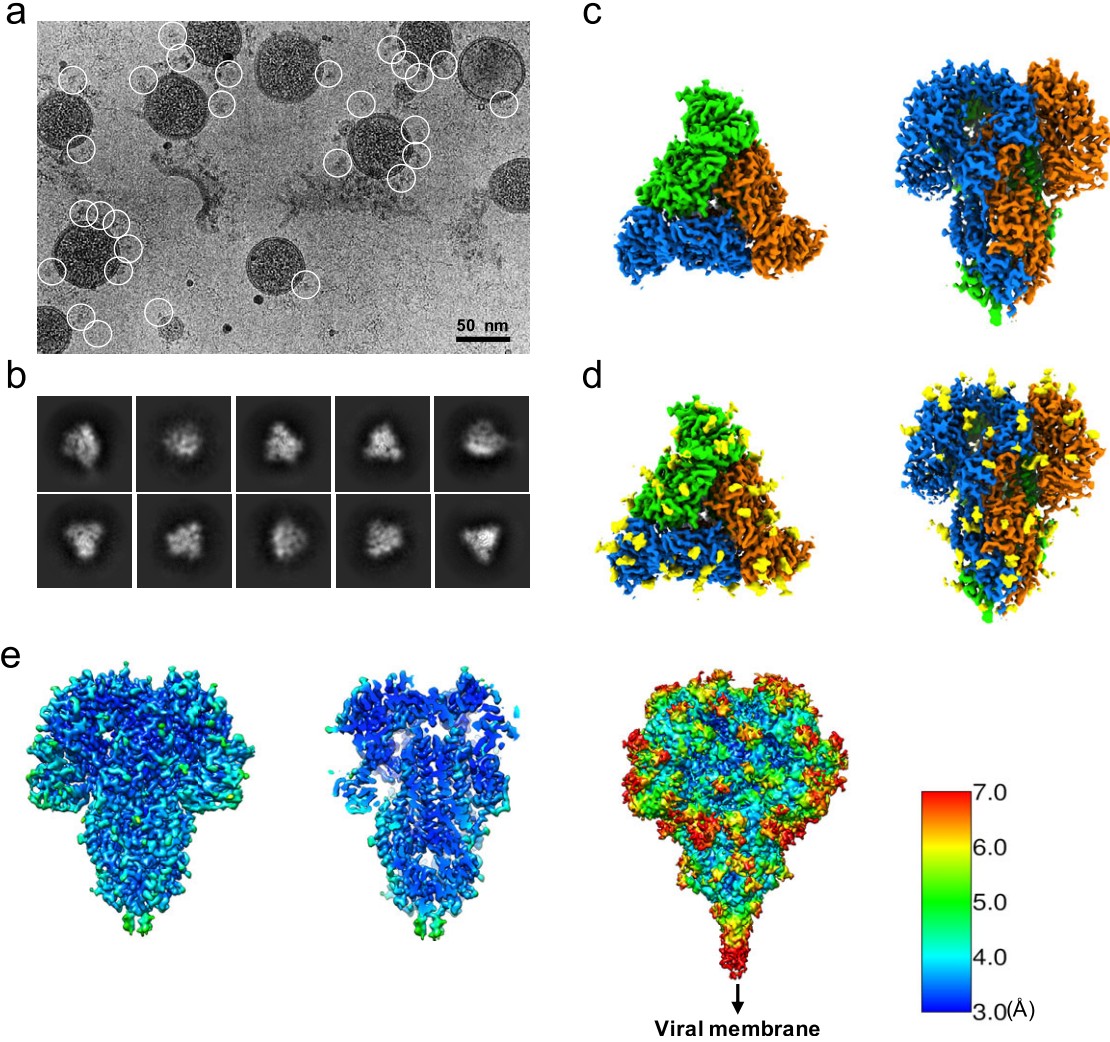

**Fig. 1.** Single-particle cryo-EM analysis of *in-situ* structure of the HCoV-NL63 coronavirus spike glycoprotein. (*a*) Representative motion-corrected cryo-EM micrograph. (*b*) Reference-free 2D class averages of computationally extracted spikes. (*c-d*) Reconstructed 3-fold symmetry-imposed cryo-EM map of the spike in the top and side views without (c) and with (d) glycans shown. (*e*) Resolution variation maps for 3D reconstruction. Left, whole map view; middle, slice view; right, whole map view at a lower threshold.

refinement was performed with and without C3 symmetry to obtain 3.4 and 3.7 Å maps, respectively (Fig. 1c–e and Fig. S1). The reconstructed map without imposed symmetry also displayed three-fold symmetry, showing a high cross-correlation coefficient (CC = 0.98) with the symmetric map (Fig. S1d). Hereafter, we will present our structural analysis of the 3.4-Å map with C3 imposed symmetry after refinement from a subset of ~82,000 S trimer particles showing adequate orientation sampling (Fig. S1). The local resolution varies in our map (Fig. 1e); the densities at the outer surface of the protein have a much lower resolution than the central region (Fig. 1e), which could be attributed to the inherent flexibility of the distal ends of glycans. Compared to the map of the purified S ectodomain (EMD-8331), our map has a more extended density without recognizable secondary structure element features that points towards the viral membrane (Fig. 1e). The detection of this feature likely results from the direct picking of S trimer anchored in the lipid envelope of native virions, which stabilizes the stem region that otherwise disassembles in the purified S ectodomains.

Because of the high feature similarity of our map to that determined from the biochemically purified S ectodomain (Walls et al., 2016), we fitted the published model (PDB ID: 5SZS) including the glycans to our 3.4-Å map and refined it. Residues 883–889 and 993–1000 that were previously unresolved were also modeled (see Methods section). The quality of the final model was validated by MolProbity (Chen et al., 2009) (Table S1), the cross correlation between the map and model (Fig. 2a), and Q-score analysis per residue (Pintilie et al., 2020) (Fig. 2b–e). The density map of our structure is better resolved in many regions, relative to the previous structure (PDB ID: 5SZS), including glycans (Fig. 3), which were previously interpreted with accompanied mass spectrometry measurements (Walls et al., 2016). Superimposition of our model and 5SZS yields 804 pruned atom pairs matched with 1.28-Å root mean square deviation (RMSD), indicating their high structural similarity. Structural comparisons based on individual domains also show similar results. These domains include domain 0, domain A, domain B (also known as RBD), domain C, and domain D, with the RMSD ranging from 0.29 Å to 0.90 Å. As shown in Fig. 1, each of the three receptor-binding domains (RBDs) within the S trimer is pointed downwards. Notably, heterogeneous refinement and 3D variability did not reveal alternative RBD conformations, implying that the native S trimer proteins on the virion are predominantly in a fully closed state, consistent with the previous report of purified HCoV-NL63 S ectodomain (Walls et al., 2016).

Glycosylation of S plays an important role in the viral life cycle and immune-evasion (Vigerust and Shepherd, 2007; Watanabe et al., 2019, 2020; Casalino et al., 2020). We observed densities protruding from the amino acid residues corresponding to all the glycan sites that were previously observed in the cryo-EM map of the ectodomain of biochemically purified S (EMD-8331). All these glycosylation sites had been identified by a combination of cryo-EM densities protruding from the expected amino acid side chains and mass spectrometry (Table 1 and Table S2) (Walls et al., 2016). The models for these glycans were fitted into our map and their Q scores are shown in Fig. 3a. Some examples are displayed as NAG residues in Fig. 3b. To further investigate the possible presence of other glycan moieties, we calculated a difference map between our map and the EMD-8331 (see Methods). We observed positive difference density in five sites shown in green in Fig. 3b. Two N-linked glycosylation sites (Asn24 and Asn203 in domain 0), predicted to exist from their sequence, as well as one O-linked glycosylation site (Ser496 in domain B), were newly identified in our difference density map (Fig. 3b and Table 1). None of these was found by

mass spectrometry or shown clearly in the previous map. These differences between our structure and the previously published one are possibly due to the differences in glycosylation patterns across species (Walls et al., 2016). In our study, the virus was grown in mammalian cells, while the previous study used S ectodomain expressed in insect cells. N-linked glycosylation in mammalian cells typically results in complex-type glycans with two to four branches extended from the tri-mannosyl core. In contrast, glycosylation in insect cells typically yields truncated, paucimannosidic or oligo-mannosidic glycans with few if any complex-type glycans (Tomiya et al., 2004). Overall, there is significant agreement between in situ and biochemically purified S protein structure with these notable differences in glycan densities. Our structure cross-validates the conformation of the spike protein of HCoV-NL63 determined from either the biochemically purified specimen or the intact virions by cryo-EM.

Compared to the S1 protein of betacoronavirus (i.e. SARS-CoV-2), HCoV-NL63 S1 has an additional N-terminal domain (domain 0) that is assumed to be a gene duplication of domain A and is a canonical feature of alphacoronaviruses. Domain B (RBD) of beta-coronaviruses transitions between upward (open) and downward (closed) conformations due to its inherent flexibility; the viral receptor ACE2 is able to sample the RBD's up conformation to initiate virus-receptor binding. In contrast, domain B of HCoV-NL63 interacts with domain A, therefore being stabilized in a closed 'circle' of HCoV-NL63 S1 (Fig. 4a). The interface area between domains A and B of HCoV-NL63 is ~500 Å$^2$ and involves 22 hydrophobic interactions (Fig. 4b) according to PDBsum structure bioinformatics analysis (Laskowski et al., 2017). The three receptor binding motifs (RBMs) on HCoV-NL63 RBD are consequently buried in the interface between domain A and domain B, and prevented from binding to the viral receptor ACE2 (Wu et al., 2009). There must exist a mechanism to induce a conformational change to release the RBMs from domain A to allow receptor binding. It was reported that HCoV-NL63 utilizes heparan sulfate proteoglycans as the attachment factor before binding to ACE2 for virus entry (Milewska et al., 2014, 2017). Whether heparan sulfate binding to HCoV-NL63 S1 or other factors induce conformational changes to release RBMs for ACE2 binding warrants further studies.

S proteins of coronaviruses require protease cleavage to release the fusion peptide that inserts into the target membrane to initiate membrane fusion for virus entry. We compared HCoV-NL63 and SARS-CoV-2 S2 fusion machineries by superimposing their structures together and an excellent agreement was observed (Fig. 4c). In both structures, the loop where the host protease cleavage site is located in the S2 region is almost perpendicular to the central helix, protruding at the periphery of the S trimer and readily accessible to host proteases. The fusion machineries of HCoV-NL63 and SARS-CoV-2 are almost identical except that the trigger loop of HCoV-NL63 forms a short alpha helix before looping back to connect to the fusion peptide while the corresponding loop of SARS-CoV-2 is long and flexible. The fusion machinery of coronaviruses provides a potential target for broad anti-coronavirus therapeutics development.

Further similarities between HCoV-NL63 and SARS-CoV-2 domains are illustrated in Fig. 5 and Fig. S2. Structure-based alignments using TM-align (Zhang and Skolnick, 2005) show that structurally, domains A and 0 from HCoV-NL63, and domain A' from SARS-CoV-2 are very similar, with RMSDs of ~3.5 Å between ~75% aligned residues. These aligned residues however have quite low sequence identities (7.5–10.2%). Regardless, due to such high structure similarity, it is very likely that they evolved from a common

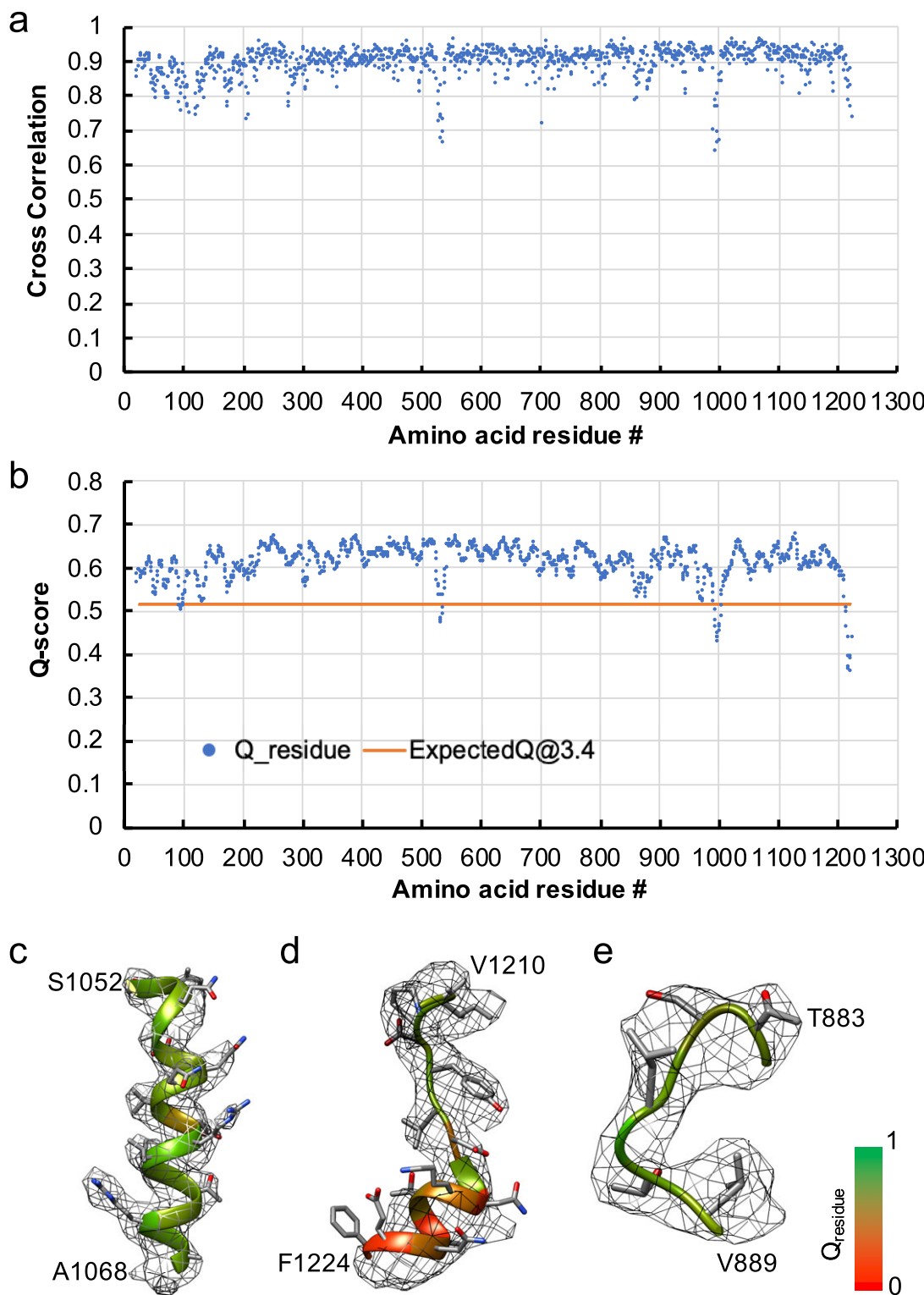

**Fig. 2.** Model validation of the HCoV-NL63 coronavirus spike glycoprotein protomer. (*a*) Per-residue cross-correlation coefficient between the model and 3.4-Å map. (*b*) Q-score for each amino acid residue in the model and 3.4-Å map; the orange line represents the expected Q-score of 0.52 at 3.4-Å resolution based on the correlation between Q-scores and map resolutions (Pintilie *et al.,* 2020). (*c–e*) Examples of different regions of the map with different resolvability: (*c*) well-resolved, (*d*) poorly-resolved; (*e*) residues not resolved in the previous biochemically purified HCoV-NL63 spike protein structure (PDB ID: 5SZS) and thus their model built *de novo* here. The model is shown as ribbon, with residue Q-scores annotated in colors. The higher Q-score indicates better resolvability.

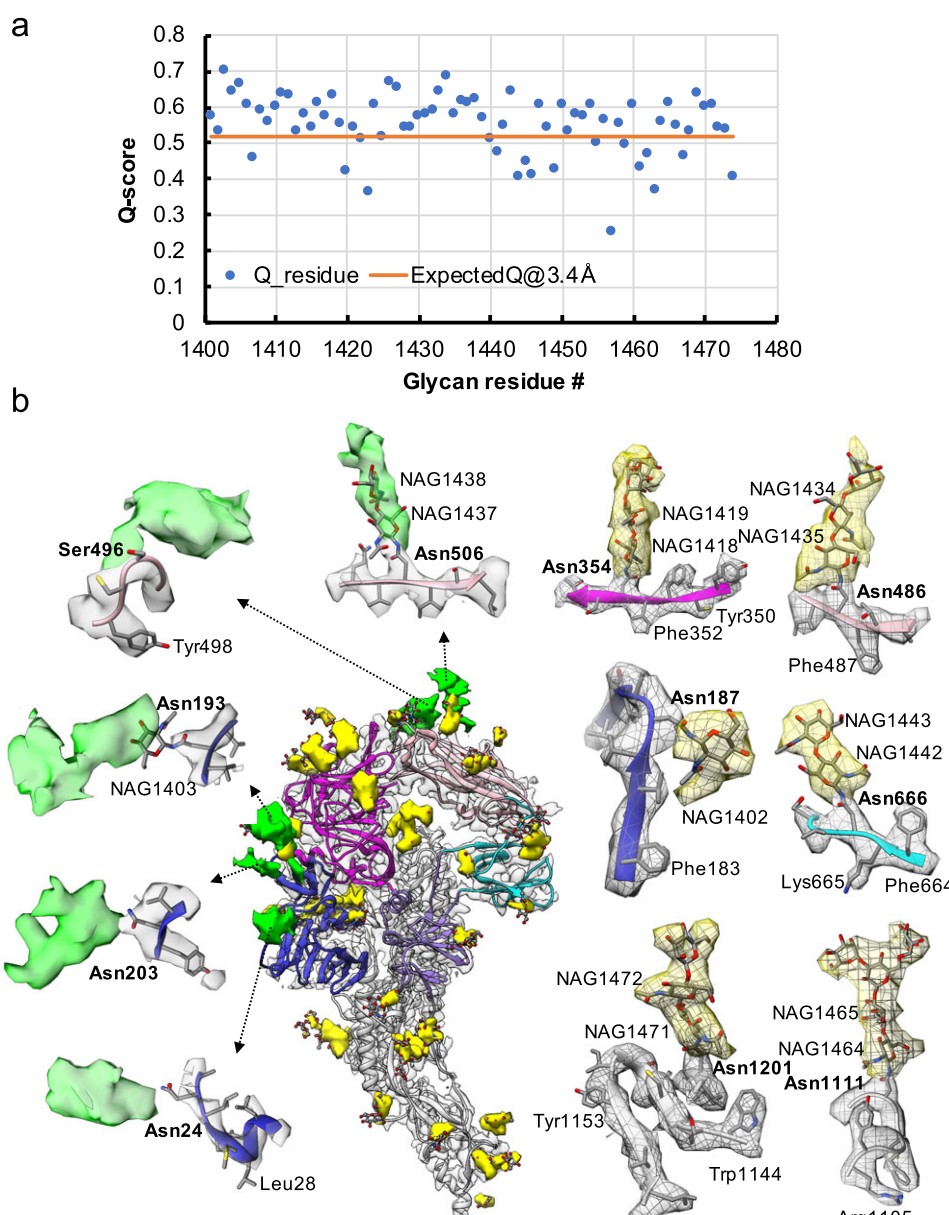

**Fig. 3.** Resolvability of glycans. (*a*) Q-scores analysis for each glycan residue (# starting from 1401) in model and map; the orange line represents the expected Q-score of 0.52 at 3.4-Å resolution based on the correlation between Q-scores and map resolutions of amino acid residues (Pintilie *et al.,* 2020). (*b*) Highlights on several glycans. Green: Positive difference density between our and previous map (EMD-8331) suggesting extra glycan densities; Yellow: glycan densities found both in our map and in the previous study. The glycan models were derived from  (PDB ID: 5SZS).

**Table 1.** Newly identified glycosylation sites in this study.

| Site | Sequon | MS identified (Walls et al., 2016) | Cryo-EM observed density (Walls et al., 2016) | Cryo-EM observed density (this study) |
|------|--------|-----------------------------------|-----------------------------------------------|---------------------------------------|
| 24 | NLSM | ND | ND | 24 |
| 203 | NYTV | ND | ND | 203 |
| 496 | GGSC | ND | ND | 496 |

Abbreviation: MS, mass spectrometry; ND, not detected; cryo-EM, cryo-electron microscopy.

ancestor, consistent with the previous hypothesis that domains 0 and A resulted from a gene duplication event of a single domain. Domains B and B′ also have a similar core fold, with RMSD of 4.13 Å, though the receptor-binding motifs look significantly different between the two, even though both bind the receptor ACE2. Domains D and D′ are also structurally similar, though still having a low sequence identity of 6.3%. Domains C/C′ and S2/S2′ are the most similar, with relatively high sequence identities of 25.4% and

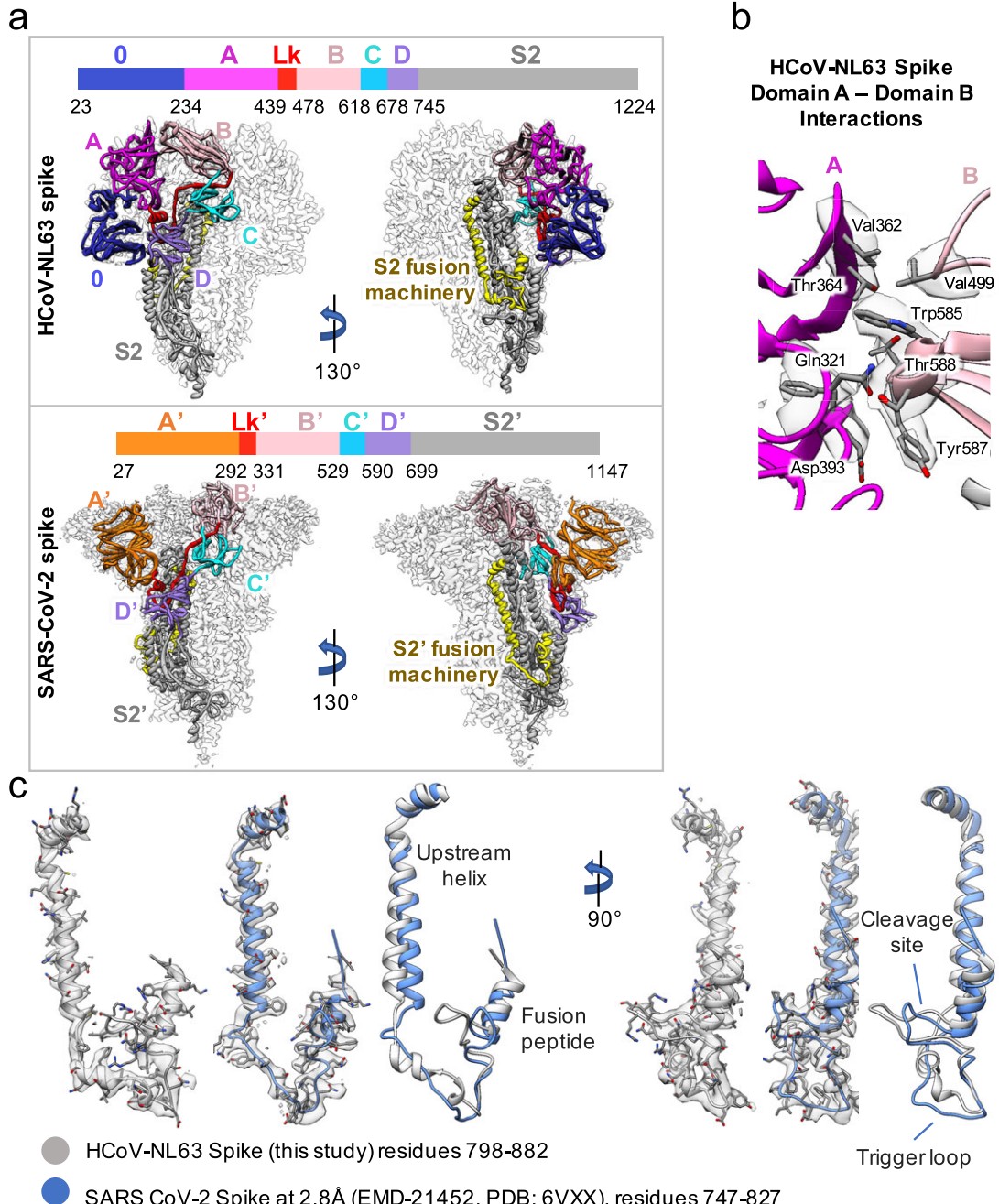

**Fig. 4.** Structure comparison between HCoV-NL63 and SARS-CoV-2 spike. (*a*) Comparison of our structure with SARS-CoV-2 spike in closed state (PDB ID: 6VXX) in two different views. Different domains and the linker (Lk) between domains A and B are indicated in different colors. (*b*) Zoom-in view to show the interactions between domain A and domain B in the HCoV-NL63 spike protein. (*c*) Extracted densities of the S2 fusion machinery region with models fitted.

37.0%; this high conservation shows that these domains, in particular, may be less prone to mutate and may be targeted by cross-reactive antibodies as reported very recently (Wang *et al.,* 2020).

The feasibility of solving the near-atomic resolution structure of membrane-anchored S trimer on the purified viral particle opens a straightforward path for follow-up structural studies of corona-viruses with external reagents like receptors, antibodies or drugs. In addition, the single-particle cryo-EM derived structures can be used together with sub-tomogram averages of the S trimer in the virion to understand complex and heterogeneous modes of interactions between S and various external reagents in different biochemical conditions, and the orientation in 3D of S with respect to the viral membrane, and, ultimately, the cell surface. Such a hybrid approach across different imaging protocols will be very useful for the discovery of authentic structural states of molecular components in a pleomorphic virus particle in the context of pathogenicity relevant to human health.

## Methods

### Cell culture and virus

Vero E6 (ATCC CRL-1586) and MA104 (ATCC CRL-2378.1) cells were maintained at 37°C in a fully humidified atmosphere with

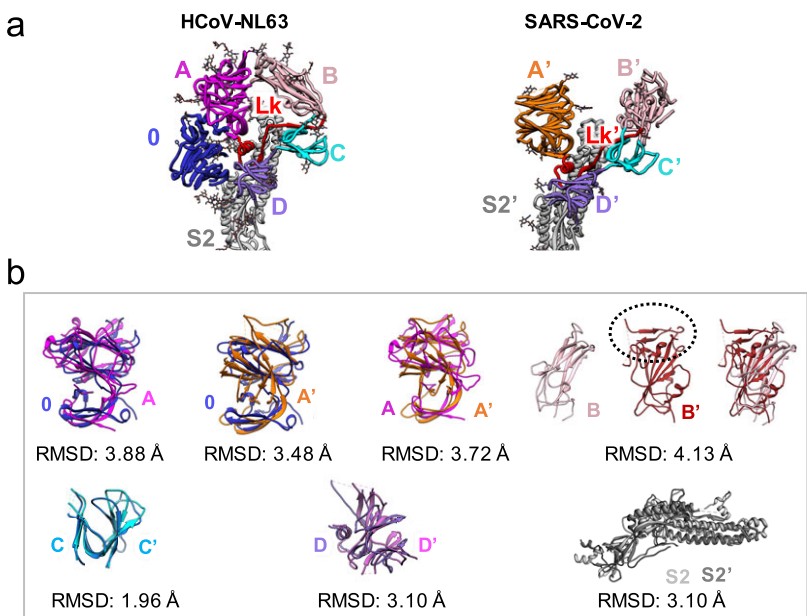

**Fig. 5.** Structural comparison of domains in HCoV-NL63 and SARS-CoV-2 spikes. (*a*) Ribbon diagram of the spikes. Each domain is shown in different colors. (*b*) Structure-based alignment of corresponding domains (PDB ID: 6VXX for A′, C′, D′ and S2′), analysed by TM-align (Zhang and Skolnick, 2005). For domain B–B′ comparison, PDB ID: 6VW1 was used for domain B′ because it has receptor-binding motif (outlined in dotted ellipse).

5% $CO_2$ in Dulbecco's Modified Eagle's medium (Invitrogen) and M199 (Gibco) medium respectively. All culture media were supplemented with penicillin and streptomycin and 10% foetal bovine serum (Hyclone). HCoV-NL63 was obtained from BEI Resources, NIAID, NIH: Human Coronavirus, NL63, NR-470.

### Virus production and purification

Monolayers of Vero E6 or MA-104 cells were infected with HCoV-NL63 at MOI 0.5. Culture supernatants were harvested when clear cytopathic effect developed. After clarification of the supernatants through a 0.45 μm filter, the virus was pelleted down through a 20% sucrose cushion. Virus was then resuspended in virus resuspension buffer (20 mM Tris, pH 8.0, 120 mM NaCl, 1 mM EDTA) before layered onto 2 ml of 60% OptiPrep (Sigma-Aldrich) and spun at 50,000*g* for 1.5 h using a SW28 rotor. After ultracentrifugation, the supernatants were removed to leave 4 ml above the virus band. The remaining 4 ml supernatant, the virus band and the underlay of 2 ml of 60% OptiPrep were mixed to reach a final concentration of 20% OptiPrep. The mixture was spun at 360,000*g* for 3.5 h with a NVT65.2 rotor. The virus band was extracted and buffer exchanged to the virus resuspension buffer using an Amicon Ultra-2 Centrifugal Filter Unit with Ultracel-100 membrane (Millipore).

### Cryo-EM vitrification, data acquisition, image processing and 3D reconstruction

Three microliters of purified HCoV-NL63 virions (4.5 mg ml$^{-1}$) were first applied to 300-mesh R2/1 + 2 nm C-film grids (Quantifoil) that had been glow-discharged for 15 s PELCO easiGlow (TED PELLA, INC.). Grids were then front-blotted for 2 s in a 90% humidified chamber and vitrified using the LEICA EM GP automated plunge freezing device. Frozen grids were then stored in liquid nitrogen until imaging. The samples were imaged in a Titan Krios cryo-electron microscope (Thermo Fisher Scientific) operated at 300 kV

with GIF energy filter (BioQuantum, Ametek) at a magnification of 64,000× (corresponding to a calibrated sampling of 1.4 Å per pixel). Micrographs were recorded by EPU software (Thermo Fisher Scientific) with a Ametek K3 Summit direct electron detector, where each image was composed of 30 individual frames with an exposure time of 3 s and an exposure rate of 16 electrons per second per Å$^2$. A total of 4,138 movie stacks were collected. All movie stacks were motion-corrected by MotionCor2 (Zheng *et al.*, 2017). Motion-corrected micrographs were imported into cryoSPARC for image processing. The contrast transfer function (CTF) was determined using the 'Patch CTF Estimation' option, and the micrographs with 'CTF fit <5 Å' were selected using the 'Manually Curate Exposures'. Then 157 particles were manually picked and subjected to 2D classification into 10 classes, 3 of which were selected as the template for the auto-particle picking, yielding 944,822 particles. Several rounds of 2D classification were then performed to remove the poor 2D class averages, and 300,236 particles were obtained. The initial map was built using ab-initio reconstruction without any symmetry applied. Next, three rounds of heterogeneous refinement were performed to further remove bad particles. The final 3D non-uniform refinement was performed using the selected 82,030 particles with or without C3 symmetry applied, and a 3.4-Å map and a 3.7-Å map were obtained, respectively. Resolution for the final maps was estimated with the 0.143 criterion of the Fourier shell correlation curve. Resolution map was calculated in cryoSPARC using the 'Local Resolution Estimation' option. The figures were prepared using UCSF Chimera (Pettersen *et al.*, 2004) or UCSF Chimera X (Goddard *et al.*, 2017).

### Model building

One protomer was first computationally extracted (Pintilie *et al.*, 2010) from the 3.4-Å cryo-EM map of the spike protein expressed on HCoV-NL63 coronavirus. The PDB coordinates of biochemically purified HCoV-NL63 spike protein (PDB ID: 5SZS) were then

fitted into the computationally extracted S protomer map. Residues 883–889 and 993–1,000 that were previously unresolved were also modeled using SWISS-MODEL (Waterhouse *et al.,* 2018). The resultant model was refined using phenix.real_space_refine (Adams *et al.,* 2010) and manually optimized with Coot (Emsley *et al.,* 2010). The atomic model of protomer was then fitted into the cryo-EM density of the other two protomers in the HCoV-NL63 spike trimer using Chimera (Pettersen *et al.,* 2004), followed by the optimization of the whole model with phenix.real_space_refine. The glycans in the PDB ID: 5SZS model were also used in our model for refinement. The final model was evaluated by MolProbity (Chen *et al.,* 2009) and Q-scores (Pintilie *et al.,* 2020). The match between the map and the model was evaluated by the correlation for each residue (Fig. 2*a*). For assessing the resolvability of the map density, the Q score was calculated for each residue with the final model (Fig. 2*b*). The Q-scores were also computed for glycan residues (Fig. 3*a*). Statistics of map reconstruction and model optimization are shown in Table S1. To identify extra glycosylation sites beyond what was previously known from the S ectodomain map (EMD-8331), we computed the difference map between our map and the previous map of the S ectodomain. When we fitted our map to this previous map using Chimera Fit in Map, it was apparent that the maps have different scales. Thus, the step size in our map was adjusted until the maximum cross-correlation between the two maps was obtained, at which point the maps also visually over-lapped better. Density values in the two maps were then normalized to an average of 0 and standard deviation of 1. The difference map was then computed at each voxel with the rescaled maps. Extra glycan densities seen in this difference map were extracted using Segger (Pintilie *et al.,* 2010) for further visualization. All figures were prepared using Chimera (Pettersen *et al.,* 2004) or Chimera X (Goddard *et al.,* 2017).

**Acknowledgements.** We thank Drs. Corey Hecksel and Patrick Mitchell for expert maintenance of Stanford-SLAC Cryo-EM Centre and the SLAC National Accelerator Laboratory for supporting conduct of these studies during a university-wide pandemic shutdown. We also thank Dr. Andriy Kryshtafovych for help with structure-based alignment analysis. This work was supported by the National Institutes of Health grants (P41GM103832, R01AI148382, P01AI120943, R01GM079429, U24GM129564 to W.C.); DOE Office of Science through the National Virtual Biotechnology Laboratory, a consortium of DOE national laboratories focused on response to COVID-19, with funding provided by the Coronavirus CARES Act.

**Conflicts of Interests.** All authors declare no competing interest.

**Authorship Contributions.** W.C. and G.S. supervised the study. J.J. prepared the sample. D.C. performed cryo-EM sample preparation. K.Z. and D.C. collected cryo-EM data. K.Z. performed cryo-EM image processing and structure determination; S.L. and K.Z. built and refined the model; K.Z., S.L., G.P., D.C., M.F.S, J.J., and W.C. analysed data. K.Z., S.L., and G.P prepared the figures. K.Z., S.L., D.C., J.J, and W.C. wrote the manuscript with input from all other authors.

**Data Deposition.** Cryo-EM maps of the HCoV-NL63 Spike protein with its associated atomic model have been deposited in the wwPDB OneDep System under EMD accession code EMD-22889 and PDB ID code 7KIP.

**Open Peer Review.** To view the open peer review materials for this article, please visit http://doi.org/10.1017/qrd.2020.16.

**Supplementary Materials.** To view supplementary material for this article, please visit http://dx.doi.org/10.1017/qrd.2020.16.

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
