## [Reviewer Report]

*Comments to Author*: The manuscript "A 3.4-Å cryo-EM structure of the human coronavirus spike trimer computationally derived from vitrified NL63 virus particles" by Zhang and co-authors reports a structure of the human coronavirus NL63 (HCoV-NL63) spike. This impactful study provides analysis of domain localisations within the HCoV-NL63 S (spike) trimer without chemical fixative in its prefusion conformation. Cryo-EM has been used in this structural study. The results of the research reported in the MS have several major outcomes. Firstly, the authors have proved the faithfulness of structural organisation of the NL63 spike obtained.Two structures (one from vitrified complete virus and another obtained using an expression system forHCoV-NL63 S ectodomain) demonstrate high level of consistency. Secondly, such consistency allowed analysing differences between the NL63 spike and the SARS-CoV-2 S structures, revealing and evaluation sites related to the receptor bindings. Thirdly, the structure of the NL63 spike from the vitrified viruses provides more complete information on glycosylation sites. This study will help to explain discrepancy in the pathogenicity between alpha and beta families of coronaviruses.

The methodologies used for the imaging and reconstruction processing followed by building the atomic modelare on the level of the state-of-the-art. The MS is well written and concise.Overall this is an important study at the current pandemic and linked to pathogenicity of the virus. It is related to the findingmedical means for suppressing infections.

It would be good if the comments listed below will be addressed:

Comparison of structures NL63 (HCoV-NL63) and SARS-CoV-2spikes indicated large rearrangements between domainsA and B. What is known about similarity (homology) of these domains and linkers between them?While a simplified schematic representation was given inWalls et all (NSMB,2016, Fig4) it does not provide sufficient information for explanation of biological relevance of this conformational differences.It would be useful to see the sequence alignment at least for domains ABCD (as a supplementary figure). Possibly, it will help to explain such big shift of the B domain in the SARS-CoV-2 S and the open conformation of the RBDs. It seems that the linker between A and B is shorter in NL63 so a different mechanism should be used for triggering the receptor binding.

It is rather difficult to find both the upstream helix and locations of the triggers loops within the entire spike structures.Therefore, it would be useful to highlight locations of the S2' trigger-loop region on side views of spikes (Figure 4a). Here may be hidden a very interesting mechanism of the spike activation.

The Fourier-Shell correlation graph should be deposited to the EMDB. Page 18 of the report: the table "Experimental information" 4 is practically empty, although these information is present in methods.

The authors have done good fitting and geometry of the model is rather good, but they did not read the report of evaluation of the fit quality:

"A red diamond above a residue indicates a poor fit to the EM map for this residue (all atom inclusion < 40%)." That effect takes place if the authors have used a too high threshold for the presentation and fitting of the model into the map. So it would make sense to try a lower threshold and the number of red diamonds will be possibly reduced and hopefully will be more consistent with other graphs in the evaluation report.

It would be good to explain why locations of ligands were poorly defined, since too many of them have high percentage of misplacements (both in bonds length and angles). While this fact does not diminish the value of the MS some verbal explanation will be helpful.

---

## [Reviewer Report]

*Comments to Author*: "A 3.4Å cryoEM structure of the human coronavirus spike trimer computationally derived from vitrified NL63 virus particles" by Zhang et al

The authors reported a 3.4Å resolution cryoEM structure of the S protein on the NL63 virus. Previously another cryoEM structure of the recombinantly expressed ectodomain of the NL63 S protein was determined to similar resolution (Walls et al., 2016). Structural comparison shows they are highly similar except that the current structure on the virus surface shows more glycosylation sites likely because the virus was grown in mammalian cells compared to the ectodomain S protein that was expressed in insect cells. Their map also showed extended density pointing towards the viral membrane although they are unable to recognize secondary structure elements. Also they showed that the RBD domain of the S proteins on the virus surface are in a "down" or "closed" state.

Glycosylation can shield the protein structures from antibody binding, therefore the level of glycosylation will affect the antigenic property of the S proteins. The results here showed that in mammalian cells, the protein are highly glycosylated. This offer important information perhaps to why antibody response is poor towards coronaviruses. The structural studies here is technically well done.

This reviewer only have minor comments:

(1)Page 4 line 113, the authors should put the reference to the map of the purified S ectodomain or its EMDB code.

(2)Page 5 line 130 "As shown in Fig. 1 and 2", I don't not see any figures regarding the RBD in a downwards conformation in Figure 2. Is that a mistake?

(3)There is no title for Table 1 and Supplementary Table S2

(4)Figure S1: It is hard to see the overlap of the two maps with and without C3 imposed.

(5)Caption of Figure 1: (c-d) should mention the cryoEM map is that of the C3 averaged map or not.

---

## [Reviewer Report]

*Comments to Author*: Reviewer #1: "A 3.4Å cryoEM structure of the human coronavirus spike trimer computationally derived from vitrified NL63 virus particles" by Zhang et al

The authors reported a 3.4Å resolution cryoEM structure of the S protein on the NL63 virus. Previously another cryoEM structure of the recombinantly expressed ectodomain of the NL63 S protein was determined to similar resolution (Walls et al., 2016). Structural comparison shows they are highly similar except that the current structure on the virus surface shows more glycosylation sites likely because the virus was grown in mammalian cells compared to the ectodomain S protein that was expressed in insect cells. Their map also showed extended density pointing towards the viral membrane although they are unable to recognize secondary structure elements. Also they showed that the RBD domain of the S proteins on the virus surface are in a "down" or "closed" state.

Glycosylation can shield the protein structures from antibody binding, therefore the level of glycosylation will affect the antigenic property of the S proteins. The results here showed that in mammalian cells, the protein are highly glycosylated. This offer important information perhaps to why antibody response is poor towards coronaviruses. The structural studies here is technically well done.

This reviewer only have minor comments:

(1)Page 4 line 113, the authors should put the reference to the map of the purified S ectodomain or its EMDB code.

(2)Page 5 line 130 "As shown in Fig. 1 and 2", I don't not see any figures regarding the RBD in a downwards conformation in Figure 2. Is that a mistake?

(3)There is no title for Table 1 and Supplementary Table S2

(4)Figure S1: It is hard to see the overlap of the two maps with and without C3 imposed.

(5)Caption of Figure 1: (c-d) should mention the cryoEM map is that of the C3 averaged map or not.

Reviewer #2: The manuscript "A 3.4-Å cryo-EM structure of the human coronavirus spike trimer computationally derived from vitrified NL63 virus particles" by Zhang and co-authors reports a structure of the human coronavirus NL63 (HCoV-NL63) spike. This impactful study provides analysis of domain localisations within the HCoV-NL63 S (spike) trimer without chemical fixative in its prefusion conformation. Cryo-EM has been used in this structural study. The results of the research reported in the MS have several major outcomes. Firstly, the authors have proved the faithfulness of structural organisation of the NL63 spike obtained.Two structures (one from vitrified complete virus and another obtained using an expression system forHCoV-NL63 S ectodomain) demonstrate high level of consistency. Secondly, such consistency allowed analysing differences between the NL63 spike and the SARS-CoV-2 S structures, revealing and evaluation sites related to the receptor bindings. Thirdly, the structure of the NL63 spike from the vitrified viruses provides more complete information on glycosylation sites. This study will help to explain discrepancy in the pathogenicity between alpha and beta families of coronaviruses.

The methodologies used for the imaging and reconstruction processing followed by building the atomic modelare on the level of the state-of-the-art. The MS is well written and concise.Overall this is an important study at the current pandemic and linked to pathogenicity of the virus. It is related to the findingmedical means for suppressing infections.

It would be good if the comments listed below will be addressed:

Comparison of structures NL63 (HCoV-NL63) and SARS-CoV-2spikes indicated large rearrangements between domainsA and B. What is known about similarity (homology) of these domains and linkers between them?While a simplified schematic representation was given inWalls et all (NSMB,2016, Fig4) it does not provide sufficient information for explanation of biological relevance of this conformational differences.It would be useful to see the sequence alignment at least for domains ABCD (as a supplementary figure). Possibly, it will help to explain such big shift of the B domain in the SARS-CoV-2 S and the open conformation of the RBDs. It seems that the linker between A and B is shorter in NL63 so a different mechanism should be used for triggering the receptor binding.

It is rather difficult to find both the upstream helix and locations of the triggers loops within the entire spike structures.Therefore, it would be useful to highlight locations of the S2' trigger-loop region on side views of spikes (Figure 4a). Here may be hidden a very interesting mechanism of the spike activation.

The Fourier-Shell correlation graph should be deposited to the EMDB. Page 18 of the report: the table "Experimental information" 4 is practically empty, although these information is present in methods.

The authors have done good fitting and geometry of the model is rather good, but they did not read the report of evaluation of the fit quality:

"A red diamond above a residue indicates a poor fit to the EM map for this residue (all atom inclusion < 40%)." That effect takes place if the authors have used a too high threshold for the presentation and fitting of the model into the map. So it would make sense to try a lower threshold and the number of red diamonds will be possibly reduced and hopefully will be more consistent with other graphs in the evaluation report.

It would be good to explain why locations of ligands were poorly defined, since too many of them have high percentage of misplacements (both in bonds length and angles). While this fact does not diminish the value of the MS some verbal explanation will be helpful.